# The Role of BDNF in Multiple Sclerosis Neuroinflammation

**DOI:** 10.3390/ijms24098447

**Published:** 2023-05-08

**Authors:** Viviana Nociti, Marina Romozzi

**Affiliations:** 1Institute of Neurology, Fondazione Policlinico Universitario Agostino Gemelli IRCCS, 00168 Rome, Italy; marinaromozzi@gmail.com; 2Centro Sclerosi Multipla, Università Cattolica del Sacro Cuore, 00168 Rome, Italy

**Keywords:** MS, brain-derived neurotrophic factor, neuroprotection, neurotrophin, CNS inflammation, demyelinating diseases

## Abstract

Multiple sclerosis (MS) is a chronic, inflammatory, and degenerative disease of the central nervous system (CNS). Inflammation is observed in all stages of MS, both within and around the lesions, and can have beneficial and detrimental effects on MS pathogenesis. A possible mechanism for the neuroprotective effect in MS involves the release of brain-derived neurotrophic factor (BDNF) by immune cells in peripheral blood and inflammatory lesions, as well as by microglia and astrocytes within the CNS. BDNF is a neurotrophic factor that plays a key role in neuroplasticity and neuronal survival. This review aims to analyze the current understanding of the role that inflammation plays in MS, including the factors that contribute to both beneficial and detrimental effects. Additionally, it explores the potential role of BDNF in MS, as it may modulate neuroinflammation and provide neuroprotection. By obtaining a deeper understanding of the intricate relationship between inflammation and BDNF, new therapeutic strategies for MS may be developed.

## 1. Introduction

Multiple sclerosis (MS) is an inflammatory and degenerative demyelinating disease of the central nervous system (CNS), which affects individuals in their early adulthood and represents the primary cause of non-traumatic disability among young and middle-aged people [1]. The etiology of MS remains unknown, but it likely involves a combination of genetic predisposition and nongenetic triggers [1]. In addition to genetic susceptibility, several environmental risk factors have been associated with an increased risk of MS. These factors include smoking, obesity during adolescence, geographical latitude, vitamin D deficiency, and Epstein–Barr virus (EBV) infection [2,3]. Recent evidence has shown that the risk of MS increased 32-fold after the infection with EBV [3].

MS can result in a range of neurologic symptoms since it can affect different regions of the brain, spinal cord, and optic nerve [4]. The term “clinically isolated syndrome” (CIS) refers to the first episode of neurological symptoms that is consistent with a demyelinating etiology and suggestive of MS. At the onset, nearly 85% of patients present with a relapsing–remitting MS (RRMS) disease type, which is characterized by episodes of acute exacerbations followed by remission [4]. According to the natural history of the disease, most RRMS patients develop a secondary progressive form of MS (SPMS), characterized by a gradual deterioration in neurological functions with or without superimposed relapses. In 5–10% of cases, patients present a primarily progressive course (PPMS), defined by a gradual accrual of disability from the beginning of the disease [4].

The pathophysiology of MS is characterized by an altered multidirectional interaction among different immune cell types in the periphery and resident CNS cells. The precise event that triggers this aberrant immune response is still not fully understood. The “outside-in” hypothesis suggests that autoreactive T cells are first activated and primed against a CNS antigen in the periphery. These T cells migrate to the CNS, where they are reactivated by antigen-presenting cells (APCs) and release cytokines that lead to direct and indirect damage to myelin [5,6].

On the other hand, the “inside-out” hypothesis involves a primary CNS degenerative process that results in the production of endogenous myelin antigens, which then trigger a secondary immune response within the CNS [7]. The debate over which of these two paradigms better represents the initial pathological processes in MS is still ongoing, although both theories could potentially be applicable [8].

The pathological hallmark of MS is multiple focal areas of demyelination, accompanied by various grades of inflammation and neurodegeneration. Pathological changes are also present in normal-appearing white matter (NAWM) and grey matter lesions [9,10,11]. The inflammatory infiltrates contain T cells, most of which are CD8+, B cells, plasma cells, and activated macrophages/microglia [10]. The classical active white matter lesion with substantial lymphocytic inflammation is characteristic of RRMS, while in progressive disease, lesions tend to have an inactive lesion core with a rim of activated macrophages and microglia [10,11,12]. In the early stages of the disease, axons are somewhat preserved; however, with the progression of MS, irreversible axonal degeneration develops [13]. Axonal damage and neuronal loss have been considered the primary causes of the progressive accumulation of irreversible disability [14].

The process of neurodegeneration has been considered to occur secondary to inflammation; however, recent findings have shown that neurodegeneration may characterize all clinical phenotypes, including in the early stages of MS. Therefore, it could be postulated that neurodegeneration is driven by independent pathological processes that may occur independently of demyelination [15,16]. To support this hypothesis, most immunomodulatory drugs, while effective in reducing clinical relapses and neuroradiological activity, have a modest influence on the development of neurodegeneration and clinical disability in the progressive disease stages [17].

Furthermore, neuropathology shows a substantial axonal loss in the NAWM, which appear to develop independently from the axonal injury in demyelinated lesions [18].

There is still a possibility that neuroinflammation continues to play a significant role in the development of MS, leading to the destruction of the myelin sheath and, ultimately, axonal damage through the release of pro-inflammatory cytokines, proteases, glutamate, and reactive oxygen species [19].

However, a significant body of evidence has shown that inflammation is not entirely detrimental. Previous studies have documented the reparative activities of the inflammatory response in neurological disorders, including MS [20]. In these conditions, immune cells produce neurotrophic factors and engage with neurons and glial cells to protect them from damage, promoting growth and repair. This phenomenon has been referred to as “protective autoimmunity” [20,21]. One possible mechanism for this neuroprotective effect is the release of neurotrophins by immune cells both in the periphery and CNS [22]. Brain-derived neurotrophic factor (BDNF) is a particularly significant neurotrophin in this context, as it may modulate neuroinflammation and provide neuroprotection in MS [23].

This review aims to provide an update on the evidence of BDNF's role in MS and its potential as a pharmacological target.

## 2. Search Strategy and Selection Criteria

We conducted a literature search using the PubMed database with search terms including “multiple sclerosis,” “pathogenesis,” “neurotrophins,” and “brain-derived neurotrophic factor”. In addition, we manually checked the bibliographies of relevant papers and reviews to locate any other pertinent articles. Finally, the extracted data from relevant papers and reviews were organized into spreadsheets.

## 3. Neuroinflammation in MS

### 3.1. MS Pathogenesis and Detrimental Effects of Neuroinflammation

#### 3.1.1. Role of the Peripheral Immune Cells

Inflammation in MS is characterized by pathogenic immune responses comprising T cells (CD4+ and CD8+ T cells), B cells, and myeloid cells through the reduced function of regulatory T cells [24].

Differentiation of CD4+ T cells results in T helper 1 (Th1), T helper 2 (Th2), or T helper 17 (Th17) cells, which are polarized in response to exposure to certain interleukins (IL) and produce specific cytokines. The cytokines secreted by Th1 cells and Th17 are proinflammatory, such as interferon gamma (IFNγ) and IL-17, while Th2 cells release anti-inflammatory cytokines, such as IL-4, IL-10, and IL-13 [25]. Differentiation of naive CD4+ T cells into the Th1 cell subset is promoted by exposure to IL-12 during the initial priming of CD4+ T cells. Th17 is another CD4+ T cell that produces a large number of cytokines (IL-17, IL-21, IL-22, and IL-26) capable of promoting inflammation [26]. IL-6 and transforming growth factor beta (TGF-β) promote the differentiation of naive CD4+ T cells into Th17 cells; IL-23 has been identified as a critical cytokine that can potentially enhance the expansion of Th17 [27]. Autoreactive Th1 and Th17 cells are thought to play a role in the development of disease and mediate the pathology of MS [28,29]. Moreover, CD4+ regulatory T cells, a unique cell subset that inhibits the function of inflammatory cells, are functionally impaired in patients with MS [30].

Upon activation, T cells can cross the blood-brain barrier (BBB) via the interaction of adhesion molecules and are reactivated by local APCs [31]. These cells, along with activated CNS-resident astrocytes and microglia, initiate a detrimental cascade that typically involves CD8+ T cells and plasma cells and causes oligodendrocyte and axonal damage through direct cell contact-dependent processes and the release of neurotoxic mediators [7].

The functional role of B cells in promoting MS neuroinflammation is also well recognized. In addition to antibody secretion by plasma cells, B cell roles include antigen presentation, production of pro-inflammatory cytokines and chemokines, and soluble toxic factors contributing to oligodendrocyte and neuronal injury and the generation of ectopic lymphoid follicles in the meninges [32,33]. These structures have a critical role in maintaining humoral immunity and an inflammatory compartmentalized response independently from peripheral inflammation, mainly in the progressive phases of the disease. Their presence correlates with the amount and size of cortical lesions, the entity of neurodegeneration, and disability [33,34,35,36].

In the late phases of MS, there is a decrease in the infiltration of immune cells into the CNS, while ongoing CNS-compartmentalized inflammation appears to influence the progressive stages [37].

#### 3.1.2. Role of CNS-Resident Cells

CNS-resident cells generally exhibit an anti-inflammatory function in the physiologic state but can switch over to a proinflammatory phenotype in inflammatory conditions such as MS [38]. Reactive astrocytes are localized in acute lesions, at the active margin of subacute lesions, and in the adjacent NAWM, suggesting their contribution to MS pathogenesis from the earliest phases of the disease [39]. Astrocytes play a role in regulating the expression of adhesion molecules and producing various neurotoxic inflammatory mediators such as cytokines, chemokines, and nitric oxide. These mediators attract both peripheral immune cells and resident CNS cells to lesion sites, leading to detrimental effects. Additionally, nitric oxide contributes to axonal mitochondrial dysfunction [38,40]. Astrocytes may play a role in inhibiting remyelination and axonal regeneration through reactive astrogliosis, glial scar formation, and the secretion of inhibitory molecules that suppress axonal growth [41]. The scar is composed mainly of astrocytes. Nonetheless, in severe lesions, there is also an interaction with other cell types, including oligodendrocyte progenitor cells and fibromeningeal cells. Reactive astrocytes have an altered phenotype and increased expression of the glial fibrillary acidic protein and other molecules such as nestin and vimentin [42]. Furthermore, astrocytes react to inflammation with hypertrophy and proliferation, ultimately resulting in an astrogliosis phenotype [42].

Similarly, microglia activation contributes to MS pathogenesis through antigen presentation, secretion of pro-inflammatory cytokines, and phagocytic processes [31,43].

Nevertheless, microglial activation in the CNS is heterogeneous and can be classified into two different subtypes: classical (M1) or alternative (M2) [44]. M1 microglia promote inflammation, while M2 microglia have an anti-inflammatory phenotype. In MS, microglia polarization is skewed towards the M1 phenotype, which plays a role in promoting inflammation and demyelination [45]. Though it seems there is a continuum of phenotypes between M1 and M2, microglia can transit from one to another [46]. The beneficial roles of microglia in MS pathogenesis will be discussed in the following paragraph.

### 3.2. The Beneficial Effect of Neuroinflammation

Although inflammation seems to have a critical role in demyelination and neurodegeneration, recent evidence suggests that inflammation is not exclusively harmful [19]. In experimental autoimmune encephalomyelitis (EAE) models, the treatment of mice with IFN-γ, classically considered a pro-inflammatory cytokine, resulted in reduced morbidity and mortality [47]. Evidence also supports the protective role of tumor necrosis factor (TNF) in EAE. Mice lacking TNF-α and its associated receptors exhibited a significant delay in remyelination [48]. TNF-α treatment dramatically reduced the disease severity in immunized mice deficient in TNF [49]. These results indicate that certain pro-inflammatory cytokines can also play a role in disease control and remyelination [47,48,49]. Anti-inflammatory cytokines such as IL-4 and IL-10 can instead have a direct protective effect [50,51].

Immune cells also exert a neuroprotective effect in MS through the production and local secretion of neurotrophins, such as nerve growth factor (NGF) and BDNF [52].

Astrocytes are key players in the pathology of MS. They may shift their response profiles over time and assume beneficial roles such as confinement of inflammation and promotion of neuroprotection [38]. Activated astrocytes have the potential to produce anti-inflammatory cytokines such as TGF-β, IL-10, and IL-27 and neurotrophic factors, including NGF and BDNF [53]. Astrocytes may also terminate the T-cell response by inducing apoptosis [38]. Astrocytes’ formation of the glial scar has historically been viewed as a detrimental process that hinders the regeneration and remyelination of axons. However, depending on the severity of the injury, the scarring process may also lead to the isolation of the inflamed area, provide structural support, and restrict damage [41].

Similarly, microglial phenotypes demonstrate temporal and spatial evolution throughout the course of MS [54]. Activated microglia can also promote remyelination by clearing myelin debris from the local environment and by secreting anti-inflammatory cytokines and several growth factors that can induce the proliferation of oligodendrocytes [54].

## 4. BDNF and Neuroinflammation

### 4.1. BDNF

BDNF is a member of the neurotrophin family, which comprises NGF, NT-3, and NT-4/5 [55]. Neurotrophins control several aspects of neuronal and oligodendroglial development and function, including differentiation, proliferation, survival, apoptosis, axonal growth, and synaptic plasticity [55]. BDNF, in particular, is one of the most largely studied neurotrophins in the mammalian brain [56].

The BDNF gene includes a common 3′-exon that encodes the pro-BDNF region and several 5′-noncoding, promoter-regulated regions, ending in a coding 5′-exon that contains the gene expression [57]. Multiple transcripts are produced due to alternative promoter usage, RNA splicing, and/or using different polyadenylation sites [57]. The synthesis and folding of the BDNF protein as a precursor form called pre-pro-BDNF occur in the endoplasmic reticulum. The pre-region sequence of the precursor is then cleaved, resulting in the production of the pro-neurotrophin isoform of BDNF (pro-BDNF). Pro-BDNF is then converted intracellularly into the mature isoform (m-BDNF) [58]. Both pro-BDNF and m-BDNF isoforms are released into the extracellular space, where pro-BDNF can be converted by metalloproteinases 2 and 9, plasmin, and extracellular proteases [59,60] (Figure 1).

Interestingly, pro-BDNF and m-BDNF have opposite effects on cellular function [61]. Pro-BDNF binds with low affinity to the p75 neurotrophin receptor (p75NTR), which is part of the TNF receptor family, and induces apoptosis. On the other hand, m-BDNF binds with high affinity to its receptor, TrkB (tyrosine kinase B), and promotes cell survival and trophic effects [62,63,64]. In the CNS, TrkB is primarily expressed in brain regions with a high degree of plasticity, such as the cortex and the hippocampus [65]. p75NTR is broadly expressed throughout the CNS during development, while in the adult brain, p75NTR is progressively downregulated, and its expression persists in limited regions, including the cholinergic basal forebrain area [66].

It is still unclear whether pro-BDNF is secreted by neurons under physiological conditions, as its level in presynaptic terminals is comparatively lower than m-BDNF [58,67].

When m-BDNF binds to the TrkB receptor, the intracellular Trk domains undergo dimerization and phosphorylation. This initiates the cytoplasmic signaling pathways, which are mediated by activation of the phosphatidylinositol 3-kinase (PI3K), mitogen-activated protein kinase (MAPK), and phospholipase C (PLC). One of the pathways leads to the hydrolysis of phosphatidylinositol bisphosphate (PIP2) by phospholipase C gamma (PLCγ), which results in the formation of IP3 and subsequent mobilization of intracellular Ca^2+^ [68]. BDNF was shown to increase Ca^2+^ levels in the cortex and hippocampus [69]. Sustained intracellular Ca^2+^ elevation activated by BDNF is thought to participate in neuronal survival and may contribute to the onset of long-term potentiation [70].

These BDNF signaling pathways activate transcription factors that condition the expression of genes encoding proteins implicated in neural plasticity and cell survival and the de novo expression of the BDNF gene [62] (Figure 2).

Neurons are the major cellular source of BDNF, which is mainly expressed in the hippocampus, cortex, amygdala, striatum, and cerebellum in rodents and humans [71]. It is also expressed and released by cells of the immune system, microglia, astrocytes, endothelial cells, and megakaryocytes [52,72,73,74]. Circulating BDNF derives from peripheral and cerebral sources because the BBB is permeable in both directions. Animal and human studies suggested a correlation between circulating and brain BDNF levels [75].

In addition to the two mentioned isoforms, the activity of BDNF is considerably impacted by the single-nucleotide polymorphism rs6265 that causes a valine (Val) to methionine (Met) substitution at codon 66 (Val66Met) in the BDNF gene pro-domain encoding region [76]. This variant can be considered a distinct ligand with independent significance for BDNF activity. It can potentially modify BDNF protein-protein interactions, binding affinities, or protein conformational stability. This polymorphism has been studied in association with several neurological conditions, including MS [76,77,78].

### 4.2. BDNF and Neuroinflammation

Recent evidence suggests that BDNF regulates inflammatory homeostasis, reducing inflammatory activity through the hedgehog and erythropoietin signaling pathways [79].

The BDNF signaling pathway also regulates the endogenous cross-talk between astrocytes and microglia in modulating neuroinflammatory mechanisms [79]. BDNF may also influence the inflammatory response by downregulating the expression of cyclooxygenase-2 (COX2) and proinflammatory cytokines in microglia (Figure 3) [79,80].

The nuclear factor kappa B (NF-κB) is a significant contributor to inflammatory activation and modulates numerous genes involved in immune and inflammatory responses, including BDNF. It also has a role in regulating inflammasome activation as well as neuronal survival, immune cell proliferation and maturation, and apoptosis [81,82]. The binding of BDNF to the TrkB receptor can also induce NF-κB expression, but the exact regulatory mechanisms of this interaction are not understood yet [83,84]. According to a recent study, the overexpression of BDNF in the hippocampus can mitigate synaptic dysfunction and improve neuroinflammation, possibly due to the inhibition of multiple signaling pathways, including the NF-κB one [80].

## 5. BDNF in MS

Studies have shown that BDNF may be involved in the pathophysiology of several neurological diseases, including MS, Alzheimer’s disease, Parkinson’s disease, and Huntington’s disease [85,86,87]. BDNF also plays a role in cerebral ischemia. In animal models, ischemic brain injury is accompanied by increased expression and levels of BDNF, which can play a role in recovery and plasticity [88]. Pretreatment with intraventricular BDNF infusion reduced infarct size after focal cerebral ischemia in rats [89]. In humans, serum BDNF levels were lower in the acute phases of ischemic stroke, suggesting that lower levels of BDNF were associated with a higher likelihood of cerebrovascular accidents [90,91].

BDNF is the most studied neurotrophin in MS for its role in the modulation of neuroinflammation and the induction of neuroprotection [23]. It has been shown to increase oligodendrocyte lineage cells and myelin proteins in rodent models, rescue injured or degenerating neurons, and stimulate axonal outgrowth [92,93].

BDNF knockout mice exhibit deficits in myelin proteins and oligodendrocyte progenitor cell proliferation [94,95]. These data favor the hypothesis that BDNF may enhance myelin repair after a demyelinating lesion. Kerschensteiner et al. demonstrated that apart from neurons and reactive astrocytes, activated T cells, B cells, and monocytes release BDNF both in vitro and in inflammatory brain lesions [52]. Moreover, TrkB, the receptor for m-BDNF, was detected in neurons near MS plaques and in reactive astrocytes within the lesion [96]. BDNF immunopositive cells also correlated with the extent of lesional demyelinating activity, being more numerous in the actively demyelinating areas than the inactive areas in the same lesion [96].

In the following sections of this review, we will summarize the main evidence from studies in serum and cerebrospinal fluid (CSF), in vivo and in vitro, of BDNF in MS and EAE models.

### 5.1. Studies on BDNF in Serum and Cerebrospinal Fluid

Some studies have reported that people with MS have lower levels of BDNF in their serum compared to healthy individuals [97,98,99,100,101]. Most of these studies focused on patients with RRMS [97,99], but in SPMS patients, the reduction in BDNF was even more pronounced [98].

Some studies also found an increase in circulating levels of BDNF during MS relapse [101,102]. These findings are consistent with the hypothesis that higher levels of BDNF are associated with active inflammation. On the contrary, other studies failed to demonstrate a significant difference in levels of BDNF in MS patients and healthy controls [103].

Studies on BDNF in the CSF of MS patients were not conclusive. A study on CSF showed that BDNF levels are reduced among RRMS patients during relapse in comparison to patients with other neurological conditions [97]. Other data demonstrated an increase in CSF BDNF levels in comparison to control subjects [104]. In SPMS patients, BDNF levels in CSF were lower compared with RRMS patients assessed during a stable phase and healthy controls [105]. Relevant studies on BDNF and MS in serum and CSF are summarized in Table 1.

### 5.2. Studies on BDNF In Vivo

BDNF has demonstrated a neuroprotective role in EAE models. This was particularly emphasized by a more aggressive disease course of EAE and increased axonal loss in mice deficient for CNS-derived BDNF in myelin oligodendrocyte glycoprotein-induced experimental EAE. Mice deficient for BDNF in immune cells displayed a weakened immune response in the early phase of EAE but progressive disability with marked axonal loss in the late stages [108]. This observation supports the hypothesis that the deficiency of immune cell-derived BDNF in the early phase of the disease impairs neuroprotection and, ultimately, enhances axonal loss in the chronic stage [108]. The supplementation of exogenous BDNF to the lesion area resulted in a less severe EAE course and direct axonal protection [108].

A study using a conditional knockout model with inducible deletion of BDNF demonstrated that the absence of BDNF during the early phase of clinical EAE led to increased clinical symptoms and structural injury. On the contrary, the deletion of BDNF in a later stage of EAE marginally influenced the clinical disease course and histopathological indicators of axonal integrity [109]. These findings imply that the neuroprotective effects of BDNF are principally significant during the early phase of EAE, suggesting that principal aspects of neurodegeneration already take place in early disease stages [109].

Chimeric mice, where BDNF was only deleted in resident cells from the CNS but not in lymphocytes, exhibited enhanced motor impairment, revealing that CNS-derived BDNF is more important than BDNF produced by immune cells for neuroprotection in EAE [109].

In a study by Makar et al. on the EAE model, treatment with BDNF delivered to the CNS via genetically engineered bone marrow stem cells delayed the clinical onset and reduced the severity of EAE, and pathological examination showed a reduction in inflammatory infiltrating cells in the brain and spinal cord lesions in comparison to control mice [110,111].

### 5.3. Studies on BDNF In Vitro

In vitro studies on BDNF and MS showed conflicting results. BDNF production by peripheral blood mononuclear cells (PBMCs) was decreased in non-treated RRMS patients when compared to healthy controls [106,107]. Therefore, the authors hypothesized that the defective regulation of BDNF secretion by immune cells and the loss of its neuroprotective activity could contribute to the deviated immune system in MS and neurodegeneration [107].

Sarchielli et al. found that the levels of BDNF in RRMS patients were not significantly different from those of healthy controls [105]. However, BDNF production in patients with RRMS was significantly higher during relapse compared with the values measured in the stable phase of the disease [105,112].

The increase in BDNF during relapses could support the hypothesis of a neuroprotective effect of BDNF produced by inflammatory cells to promote survival and remyelination through the production of neurotrophins [105].

Significantly lower BDNF levels were found in the PBMCs of patients with SPMS compared to healthy controls. Furthermore, this reduction was more pronounced in a subgroup of patients with greater deterioration in disability [105]. The authors speculated that the reduced levels of BDNF in the PBMCs of patients with SPMS may contribute to neurodegeneration and the progression of the disease [105].

RRMS patients with active lesions had increased basal and stimulated levels of BDNF compared to those without [105,113]. Recent MRI studies also confirmed the link between BDNF production and the inflammatory response. In addition, a significant positive correlation arose between the number of gadolinium-enhancing lesions and BDNF levels [105,114].

Weinstock-Guttman et al. discovered that immune cell BDNF production was associated with increased inflammatory activity in the white matter and with microscopic damage in the NAWM in MS patients, and they suggested that BDNF secretion from immune cells may be implicated in the initial stages of MS pathogenesis [114].

On the contrary, other studies have reported that BDNF production by PBMCs is higher in RRMS patients compared to controls [115].

Nevertheless, a recent study found elevated levels of pro-BDNF in immune cells, suggesting a possible role for the pro-neurotrophin isoform in the pathogenesis of MS [116].

### 5.4. Disease Modifying Therapies and BDNF

The effect of disease-modifying therapies (DMTs) on the concentrations of BDNF in MS patients has also been studied. Glatiramer acetate (GA), a synthetic amino acid copolymer whose effects are thought to be primarily mediated by a Th1 to Th2 shift of GA-reactive T lymphocytes, has been demonstrated to induce BDNF in GA-reactive T cells [117,118]. These data indicate that GA-reactive T cells are a critical source of BDNF and may be involved in neuroprotection. The anti-inflammatory and neuroprotective properties of GA have been seen in responder MS patients, with a decrease in serum proinflammatory cytokines and an increase in BDNF levels in PBMCs [97,119]. A study by Azoulay et al. showed higher serum BDNF levels in GA-treated MS patients in comparison to untreated patients, at levels similar to those seen in healthy individuals [97].

However, conflicting findings emerge from the research on interferon beta (IFN-β) and BDNF. In vitro studies reported no effect of IFN-β on BDNF levels [113,115], while others showed an IFN-β-dependent increase in BDNF concentration [106,120].

Fingolimod, a drug targeting sphingosine-1 phosphate receptors, has been shown to increase the levels of BDNF in rodent neurons [121].

## 6. Therapeutical Implications

Currently, available treatments for MS are limited to immunomodulatory therapies that aim to lower disease activity and slow its progression. At present, none of the treatments for RRMS can prevent axonal loss and neurodegeneration, which represent the major contributors to irreversible clinical disability in MS patients. This concept has received considerable attention in the past decade, and therapeutic promotion of remyelination seems to be an appealing option for preventing MS progression [122]. Therefore, increasing BDNF levels may be a potential therapeutic strategy for MS. When considering the use of growth factors to treat MS and other neurological diseases, challenges arise in increasing their levels in specific regions of the CNS and enabling their penetration from the periphery through the BBB. Evidence suggests that growth factors do not readily cross the BBB in vivo [123].

Huang and Dreyfus reviewed all the possible strategies to allow the penetration of growth factors in the CSN, including the introduction of growth factors by injection of viral vectors or encapsulated engineered cells, the use of intranasal delivery or mesenchymal stem cells, or conjugating the growth factor with nanoparticles able to cross the BBB [124]. Recently, there has been a study on the possibility of delivering BDNF into the CNS through a BBB modulator called ADTC5 in EAE models. The administration of BDNF and ADTC5 significantly improved the clinical performance of EAE mice and induced remyelination [125]. More research is needed to confirm these results and find a non-invasive system to deliver BDNF to the CNS via systemic administration. Finally, multiple DMTs seem to exert their effects, in part, through the action of growth factors, including BDNF [117,118].

## 7. Conclusions

The concept that CNS inflammation is invariably detrimental in EAE/MS has been disputed in the last decade, and there is now strong evidence supporting a protective counter mechanism induced by inflammation that aims to reestablish the immune-privileged status of the CNS soon after the pathogenic event. Neurotrophins are now regarded as crucial players in the pathogenetic scenario of CNS immune inflammation. BDNF, in particular, seems to be a crucial regulator of neuroinflammatory mechanisms and of myelin repair after immune system-mediated damage [79,94]. Nevertheless, there is no complete understanding of the exact role of BDNF in MS, and studies on the topic are often inconclusive. Some possible explanations for these conflicting results could be differences in the techniques used to measure BDNF levels in MS patients across different studies. These methods can differ in their sensitivity, specificity, and accuracy, affecting the measurements’ precision and reliability. Furthermore, the selection of the study population, the stage and severity of the disease, and the presence of comorbidities may also affect BDNF measurements. Similarly, the use of different treatments may also affect BDNF levels in MS patients. Therefore, the variations in BDNF measurements across different studies may reflect differences in the study design and sample selection. Future studies should aim to standardize the methods for measuring BDNF levels in MS patients to improve the consistency and comparability of the results.

However, comprehending the relationship between inflammation and neurodegeneration is essential for developing new, effective therapeutic approaches. This is because the mainstay of treatment for MS is moving towards preventing neurodegenerative phenomena, slowing disease progression, and providing neuroprotection. Therefore, strategies that increase BDNF levels or enhance its function may be promising therapeutic options for this disease.

## Figures and Tables

**Figure 1 ijms-24-08447-f001:**
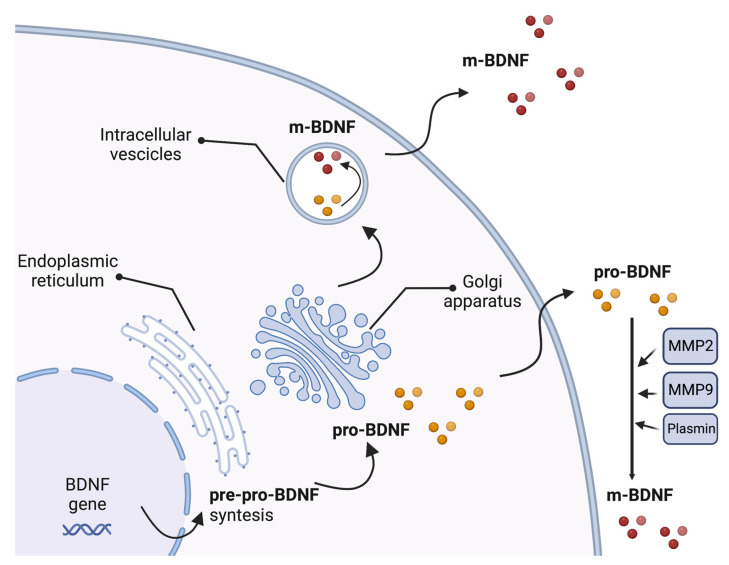
Production of brain-derived neurotrophic factor (BDNF). The synthesis and folding of the BDNF protein in a precursor form called pre-pro-BDNF occur in the endoplasmic reticulum. The pre-region sequence of the precursor is then cleaved, resulting in the production of the pro-neurotrophin isoform of BDNF (pro-BDNF). Following cleavage of the pro-domain sequence, pro-BDNF is then converted into the mature isoform (m-BDNF). Both pro-BDNF and m-BDNF isoforms are released into the extracellular space, where pro-BDNF can be converted by metalloproteinases 2 (MMP2) and 9 (MMP9), plasmin, and extracellular proteases. BDNF, brain-derived neurotrophic factor; m-BDNF, the mature isoform of BDNF; MMP2, metalloprotease 2; MMP9, metalloprotease 9; pre-pro-BDNF, the uncleaved precursor form of BDNF; pro-BDNF, the pro-neurotrophin isoform of BDNF.

**Figure 2 ijms-24-08447-f002:**
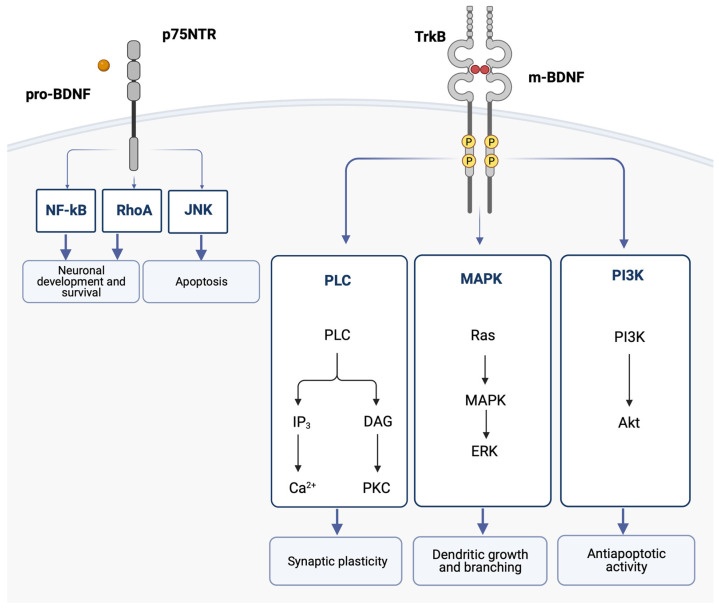
The binding of the mature BDNF isoform (m-BDNF) to the tyrosine kinase B (TrkB) receptor leads to dimerization and phosphorylations in the intracellular Trk domains, thereby initiating the cytoplasmic signaling pathways mediated by activation of the phosphatidylinositol 3-kinase (PI3K), the mitogen-activated protein kinase (MAPK), and phospholipase C (PLC). Ultimately, the activation of these pathways leads to dendritic growth and branching, as well as regulation of apoptosis and synaptic plasticity. The binding of the pro-neurotrophin isoform of BDNF (pro-BDNF) to the p75 neurotrophin receptor (p75NTR) leads to the activation of the c-Jun N-terminal kinases (JNK) signaling pathway, which promotes apoptosis, and the RhoA (Ras homolog gene family member) and nuclear factor kappa B (NF-ĸB) signaling pathways, which cause neuronal growth and neuronal survival. Akt, protein kinase B; BDNF, brain-derived neurotrophic factor; DAG, diacylglycerol; ERK, extracellular signal-regulated kinases; IP3, inositol trisphosphate 3; JNK, c-Jun N-terminal kinases; MAPK, mitogen-activated protein kinase; NF-ĸB, nuclear factor kappa B; PI3K, phosphatidylinositol 3-kinase; PCK, protein kinase C; PLC, phospholipase C; RhoA, Ras homolog gene family member; TrkB, tyrosine kinase B.

**Figure 3 ijms-24-08447-f003:**
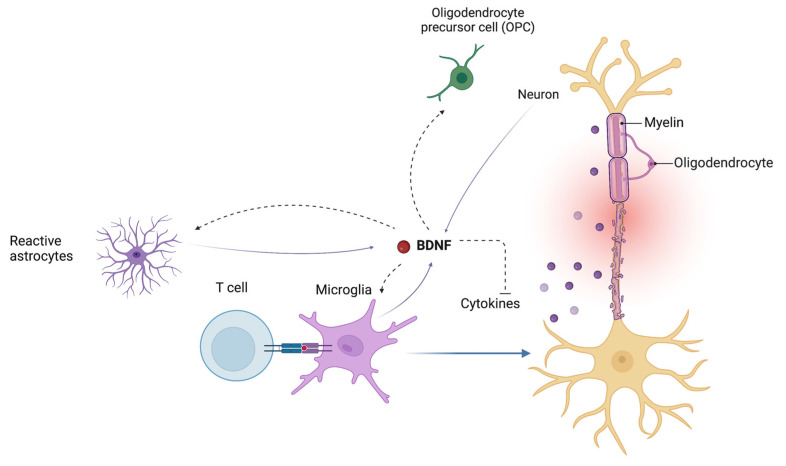
Brain-derived neurotrophic factor (BDNF) influences the endogenous cross-talk between astrocytes and microglia in modulating neuroinflammatory mechanisms. BDNF plays an important role in the regulation of the proliferation and differentiation of oligodendrocyte precursor cells (OPCs) into mature oligodendrocytes. BDNF, brain-derived neurotrophic factor; OPC, oligodendrocyte precursor cells.

**Table 1 ijms-24-08447-t001:** Relevant studies on BDNF and MS in serum and cerebrospinal fluid (CSF).

Author (Date)	Type of Sample	BDNF Levels	Study Population	Main Findings
Azoulay (2005) [97]	Serum	↓	74 RRMS, 28 HC	BDNF levels were lower in untreated RRMS patients than in HC
Naegelin (2020) [98]	Serum	↓	259 pwMS (11 CIS, 178 RRMS, 56 SPMS, 14 PPMS), 259 HC	BDNF levels were lower in pwMS than in HC. BDNF levels in patients with SPMS were lower than in RRMS
Wens (2016) [99]	Serum	↓	22 RRMS, 19 HC	BDNF levels were lower in RRMS patients than in HC
Castellano (2007) [100]	Serum	↓	11 RRMS, 11 HC	BDNF levels were lower in RRMS patients than in HC
Frota (2009) [101]	Serum	↓	29 RRMS, 24 HC	BDNF levels were lower in RRMS patients than in HC, BDNF levels increased significantly after MS relapse
Oraby (2021) [102]	Serum	=	60 RRMS, 30 HC	No difference in BDNF levels between patients in relapse or remission and HC BDNF levels were higher in MS patients in relapse than in remission
Damasceno (2015) [103]	Serum	=	21 RRMS, 9 HC	No difference in BDNF levels between RRMS patients and HC
Lalive (2008) [106]	Serum	=	30 RRMS, 15 HC	No difference in BDNF levels between RRMS untreated patients and HC
Sarchielli (2002) [105]	CSF	↓	35 pwMS (20 RRMS, 15 SPMS), 20 HC	BDNF levels were lower in SPMS patients than in RRMS patients, and HC
Mashayekhi (2009) [104]	CSF	↑	48 RRMS, 53 HC	BDNF levels were higher in RRMS patients than in HC
Azoulay (2005) [107]	CSF	↓	9 RRMS, 7 controls	BDNF levels were lower in untreated RRMS patients than in patients with other neurological diseases

BDNF, brain-derived neurotrophic factor; CIS, clinically isolated syndrome; CSF, cerebrospinal fluid; HC, healthy controls; MS, multiple sclerosis; PPMS, primary-progressive MS; PwMS, people with multiple sclerosis; RRMS, relapsing-remitting MS; SPMS, secondary-progressive MS.

## Data Availability

Not applicable.

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
