# Peer review of "The Role of BDNF in Multiple Sclerosis Neuroinflammation"

_ijms, 2023, doi:10.3390/ijms24098447_

Round 1

Reviewer 1 Report

Dear Authors,

I read with interest your review manuscript on the role of BDNF in neuroinflammation linked to multiple sclerosis.

Before recommending the manuscript for publicatio in IJMS, I require the following points to be addressed:

MAJOR points

The review is well written, but I think that the main focus, i.e. BDNF and neuroinflammation in MS, should be expanded. On the other hand, the introductory part (§3) on neuroinflammation in MS could be shortened and amended taking into account my specific comments (please see below).

With respect to expanding the main focus of the manuscript, I think that the following papers should be cited in the places Authors deem most appropriate:

10.1155/2022/3002304

10.3390/ijms232012183

10.3390/genes13020332

10.3389/fimmu.2021.792465.

10.3390/ijms22179495

10.3390/biom11040504

10.1016/j.jneuroim.2018.02.016

10.1016/j.jneuroim.2016.01.002

10.1016/j.jneuroim.2012.07.014

10.1016/j.jneuroim.2004.10.026

10.1016/j.intimp.2022.109647

10.7150/thno.51390

10.1111/cns.13111

10.1016/j.jneuroim.2018.12.010

10.1016/j.yexmp.2015.06.016

- §3.1 introduces some general immunological processes, with quick links to MS. However, it fails to review the specific cellular-molecular processes and inflammatory mediators (e.g., cytokines) involved in MS pathogenesis.

This § should be rewritten to provide more specific details on the relationship between neuroinflammation and MS. Or, better, I suggest splitting it into a § dealing with non-CNS immune cells and a § reviewing evidence for CNS-driven (astrocytes, microglia) inflammatory and neurodegenerative processes.

- Lines 51-2: please provide references.

- Lines 97-103: What are the features of this general immunological mechanism that are specific to MS?

- Lines 107-9: please provide reference(s)

- Lines 112-8: What are the features of reactive astrogliosis that are specific to MS?

- Lines 159-63: microglial phenotypes should be reviewed more in detail. The ongoing debate about a redefinition of microglial activation classes should be taken into account here.

- §4.3: Reviewing the role of BDNF in neurological diseases is not the primary focus of this manuscript. Thus, I suggest removing this § and merging its few sentences as an introductory part for §5.

- §5.3 should be re-titled, as analyses on blood and CSF are not ex vivo studies (e.g., experiments on acute or organotypic brain slices).

- Table 1: to improve immediacy of read and comparison among different studies, I suggest using graphics (e.g. up- or downward-pointing arrows) and a "BDNF level" column, flanked by columns specifying the type of assay (e.g., blood, CSF...) and experimental/study groups.

MINOR points

- Line 3: please add affiliations for both Authors

- Lines 17-8: keywords should not be also contained in the title

- Lines 24-6: Even though this review manuscript does not focus on the aetiology of MS, I suggest spending something more articulated that a short sentence on the possible risk factors, also in light of the hype generated by recent studies on the possible role of EBV infection.

- Line 50: "NAWM" should be "NAWMs"

- Line 52: "activate" should be "activated", I think.

- Lines 67-8: I suggest changing "the NAWM" into "NAWMs" and "appears" into "appear".

- Line 82: "BDNF's" should be "BDNF".

- §s 3 and 4 have the same title.

- §4.1: a brief outline of the different BDNF mRNA isoforms would benefit the Reader.

§4.1: please also outline the expression patterns of TrkB and p75NTR in the adult brain.

Author Response

Reviewer #1 reports:

The review is well written, but I think that the main focus, i.e. BDNF and neuroinflammation in MS, should be expanded. On the other hand, the introductory part (§3) on neuroinflammation in MS could be shortened and amended taking into account my specific comments (please see below).

With respect to expanding the main focus of the manuscript, I think that the following papers should be cited in the places Authors deem most appropriate:

10.1155/2022/3002304

10.3390/ijms232012183

10.3390/genes13020332

10.3389/fimmu.2021.792465.

10.3390/ijms22179495

10.3390/biom11040504

10.1016/j.jneuroim.2018.02.016

10.1016/j.jneuroim.2016.01.002

10.1016/j.jneuroim.2012.07.014

10.1016/j.jneuroim.2004.10.026

10.1016/j.intimp.2022.109647

10.7150/thno.51390

10.1111/cns.13111

10.1016/j.jneuroim.2018.12.010

10.1016/j.yexmp.2015.06.016

We thank the referee for his/her advice, and we have added the following references to the manuscript:

- 10.7150/thno.51390

-10.1016/j.jneuroim.2004.10.026

- 10.1016/j.jneuroim.2012.07.014

- 10.3390/genes13020332

- §3.1 introduces some general immunological processes, with quick links to MS. However, it fails to review the specific cellular-molecular processes and inflammatory mediators (e.g., cytokines) involved in MS pathogenesis.

We thank the Referee for her/his important suggestion. Accordingly, we modified the manuscript by adding the following sentence “Differentiation of naive CD4+ T cells into the Th1 cell subset is promoted by exposure to IL-12 during the initial priming of CD4+ T cells. Th17 is another CD4+ T cell that induces a large number of cytokines (IL-17, IL-21, IL-22 and IL-26) capable of promoting inflammation. IL-6 and transforming growth factor beta (TGF-β) promote the differentiation of naive CD4+  T cells into Th17 cells; IL-23 has been identified as a critical cytokine that can potentially enhance the expansion of Th17”.

This § should be rewritten to provide more specific details on the relationship between neuroinflammation and MS. Or, better, I suggest splitting it into a § dealing with non-CNS immune cells and a § reviewing evidence for CNS-driven (astrocytes, microglia) inflammatory and neurodegenerative processes.

We thank the referee for his/her advice, and accordingly, we added a subparagraph in section 3.1.2 entitled “Role of CNS-resident cells”.

- Lines 51-2: please provide references.

We thank the referee, and we accordingly added the following reference “Popescu, B.F.; Pirko, I.; Lucchinetti, C.F. Pathology of multiple sclerosis: where do we stand? Continuum (Minneap Minn) 2013, 19, 901-921, doi:10.1212/01.Con.0000433291.23091.65”:

- Lines 97-103: What are the features of this general immunological mechanism that are specific to MS?

We thank the referee for highlighting this important point, and accordingly, we added the following sentence “Autoreactive Th1 and Th17 cells are thought to play a role in the development of disease and mediate the pathology of MS. Moreover, CD4+ regulatory T (Treg) cells, a unique cell subset that inhibits the function of inflammatory cells, are functionally impaired in patients with MS”.

- Lines 107-9: please provide reference(s)

We thank the Referee for her/his important suggestion, and we added the following reference: “Correale, J.; Farez, M.F. The Role of Astrocytes in Multiple Sclerosis Progression. Front Neurol 2015, 6, 180, doi:10.3389/fneur.2015.00180”.

- Lines 112-8: What are the features of reactive astrogliosis that are specific to MS?

We thank the referee for raising this crucial point. We clarified the features of reactive astrogliosis by adding the following sentence “The scar is composed mainly of astrocytes. Nevertheless, in severe lesions, there is also an interaction with other cell types, including oligodendrocyte progenitor cells and fibromeningeal cells. Reactive astrocytes have an altered phenotype and increased expression of the glial fibrillary acidic protein (GFAP) and other molecules such as nestin and vimentin. Furthermore, astrocytes react to inflammation with hypertrophy and proliferation, ultimately resulting in an astrogliosis phenotype”.

- Lines 159-63: microglial phenotypes should be reviewed more in detail. The ongoing debate about a redefinition of microglial activation classes should be taken into account here.

We thank the referee to raise this crucial point. And we have added the following sentence: “Nevertheless, microglial activation in the CNS is heterogeneous and can be classified into two different subtypes: classical (M1) or alternative (M2). M1 microglia promotes inflammation, while M2 microglia is the anti-inflammatory phenotype. In MS, microglia polarization is skewed towards the M1 phenotype, which plays a role in promoting inflammation and demyelinating. Though it seems there is a continuum of phenotypes between M1 and M2, and microglia can transit from one to another. The beneficial roles of microglia in MS pathogenesis will be discussed in the following paragraph”.

- §4.3: Reviewing the role of BDNF in neurological diseases is not the primary focus of this manuscript. Thus, I suggest removing this § and merging its few sentences as an introductory part for §5.

We thank the referee, and we completely agree with the suggestion given. Accordingly, we deleted the paragraph and left a sentence on the topic “Studies have shown that BDNF may be involved in the pathophysiology of several neurological diseases, including MS, Alzheimer's disease, Parkinson's disease and Huntington's disease. BDNF also plays a role in cerebral ischemia. In animal models, ischemic brain injury is accompanied by increased expression and levels of BDNF, which can play a role in recovery and plasticity. Pretreatment with intraventricular BDNF infusion showed to reduce infarct size after focal cerebral ischemia in rats. In humans, serum BDNF levels were lower in the acute phases of ischemic stroke, suggesting that lower levels of BDNF were associated with a higher likelihood of cerebrovascular accidents”.

- §5.3 should be re-titled, as analyses on blood and CSF are not ex vivo studies (e.g., experiments on acute or organotypic brain slices).

We thank the referee for his/her suggestion, and accordingly we have changed the title to “Studies on BDNF in serum and cerebrospinal fluid”.

- Table 1: to improve immediacy of read and comparison among different studies, I suggest using graphics (e.g. up- or downward-pointing arrows) and a "BDNF level" column, flanked by columns specifying the type of assay (e.g., blood, CSF...) and experimental/study groups.

We thank the referee for this suggestion, and accordingly we have made the changes suggested in Table 1.

MINOR points

- Line 3: please add affiliations for both Authors

We apologize for the mistake, and accordingly, we added the affiliation.

- Lines 17-8: keywords should not be also contained in the title

We apologize for the mistake, and accordingly, we modified the keywords as follows: MS; brain-derived neurotrophic factor; neuroprotection; neurotrophin; CNS inflammation; demyelinating diseases. 

- Lines 24-6: Even though this review manuscript does not focus on the aetiology of MS, I suggest spending something more articulated that a short sentence on the possible risk factors, also in light of the hype generated by recent studies on the possible role of EBV infection.

We thank the referee for his/her suggestions, and accordingly, we added the following sentence and respective references “Besides genetic susceptibility, several environmental risk factors have been associated with an increased risk of MS, such as smoking, obesity during adolescence, geographical latitude, vitamin-D deficiency and Epstein-Barr virus (EBV) infection. Recent evidence has shown that the risk of MS increased 32-fold after the infection with EBV”.

- Line 50: "NAWM" should be "NAWMs"

We thank the referee, and we modified it as suggested.

- Line 52: "activate" should be "activated", I think.

We thank the referee, and we accordingly corrected it.

- Lines 67-8: I suggest changing "the NAWM" into "NAWMs" and "appears" into "appear".

We thank the referee, and we modified it as suggested.

- Line 82: "BDNF's" should be "BDNF".

We thank the referee, and we modified it as suggested.

- §s 3 and 4 have the same title.

We thank the referee, and we substituted it with the following title “BDNF and neuroinflammation”.

- §4.1: a brief outline of the different BDNF mRNA isoforms would benefit the Reader.

We thank the referee, and we added the following sentence “The BDNF gene includes a common 3′-exon that encodes the pro-BDNF region, and several 5′-noncoding, promoter-regulated regions, ending in a coding 5′-exon that contains the gene expression. Multiple transcripts are produced due to alternative promoter usage, RNA splicing, and/or using different polyadenylation sites”.

- 4.1: please also outline the expression patterns of TrkB and p75NTR in the adult brain.

We thank the referee for his/her advice, and we added the following sentence “In CNS, TrkB is primarily expressed in brain regions with a high degree of plasticity, such as the cortex and the hippocampus; p75NTR is broadly expressed throughout the CNS during the development, while in the adult brain p75NTR is progressively downregulated, and its expression persists in limited regions including the cholinergic basal forebrain area”.

Reviewer 2 Report

The manuscript "The role of BDNF in multiple sclerosis neuroinflammation" by V. Nociti and M. Romozzi is review about the  significance of neurotrophic factor BDNF in MS neuroiflammation and its potential use in new therapeutic strategies for MS.

The manuscript is well organized and the content is in accordance with the title. The results of the studies with BDNF levels in serum and CSF are clearly presented, as well as results of different BDNF research models. The conclusions point to the problems regarding BDNF measurements, and consequent lack of understending of its role. The references are appropriate but 69 of 107 references are older than 10 years. I believe that some more recent can be added.

Considering all the above, I believe that the manuscript will be interesting and useful to the scientific public of the Journal.I suggest acceptance of the manuscript after a minor revision related to the update of references.

Author Response

Reviewer #2 reports:

The manuscript "The role of BDNF in multiple sclerosis neuroinflammation" by V. Nociti and M. Romozzi is review about the significance of neurotrophic factor BDNF in MS neuroiflammation and its potential use in new therapeutic strategies for MS.

The manuscript is well organized and the content is in accordance with the title. The results of the studies with BDNF levels in serum and CSF are clearly presented, as well as results of different BDNF research models. The conclusions point to the problems regarding BDNF measurements, and consequent lack of understending of its role. The references are appropriate but 69 of 107 references are older than 10 years. I believe that some more recent can be added.

Considering all the above, I believe that the manuscript will be interesting and useful to the scientific public of the Journal. I suggest acceptance of the manuscript after a minor revision related to the update of references.

We thank the Referee for his/her comment on our review and for highlighting the crucial point of updating the references with more recent works. Accordingly, we added the following references:

  • Guo, S.; Wang, H.; Yin, Y. Microglia Polarization From M1 to M2 in Neurodegenerative Diseases. Frontiers in Aging Neuroscience 2022, 14, doi:10.3389/fnagi.2022.815347.
  • Tang, Y.; Le, W. Differential Roles of M1 and M2 Microglia in Neurodegenerative Diseases. Mol Neurobiol 2016, 53, 1181-1194, doi:10.1007/s12035-014-9070-5.
  • Kunkl, M.; Frascolla, S.; Amormino, C.; Volpe, E.; Tuosto, L. T Helper Cells: The Modulators of Inflammation in Multiple Sclerosis. Cells 2020, 9, doi:10.3390/cells9020482.
  • Prajeeth, C.K.; Kronisch, J.; Khorooshi, R.; Knier, B.; Toft-Hansen, H.; Gudi, V.; Floess, S.; Huehn, J.; Owens, T.; Korn, T., et al. Effectors of Th1 and Th17 cells act on astrocytes and augment their neuroinflammatory properties. Journal of Neuroinflammation 2017, 14, 204, doi:10.1186/s12974-017-0978-3.
  • Chaturvedi, P.; Singh, A.K.; Tiwari, V.; Thacker, A.K. Brain-derived neurotrophic factor levels in acute stroke and its clinical implications. Brain Circ 2020, 6, 185-190, doi:10.4103/bc.bc_23_20.
  • Bjornevik K, Cortese M, Healy BC, Kuhle J, Mina MJ, Leng Y, Elledge SJ, Niebuhr DW, Scher AI, Munger KL, Ascherio A. Longitudinal analysis reveals high prevalence of Epstein-Barr virus associated with multiple sclerosis. Science. 2022 Jan 21;375(6578):296-301. doi: 10.1126/science.abj8222.
  • Dolcetti, E.; Bruno, A.; Azzolini, F.; Gilio, L.; Moscatelli, A.; De Vito, F.; Pavone, L.; Iezzi, E.; Gambardella, S.; Giardina, E., et al. The BDNF Val66Met Polymorphism (rs6265) Modulates Inflammation and Neurodegeneration in the Early Phases of Multiple Sclerosis. Genes 2022, 13, 332.
  • Goverman, J.M. Regulatory T Cells in Multiple Sclerosis. New England Journal of Medicine 2021, 384, 578-580, doi:10.1056/NEJMcibr2033544.

Reviewer 3 Report

1) Line 137 - sentence broken

2) Express and are able to release BDNF from neurons

3) Figure 2 should also reflect the mechanism of action of pro-BDNF

4) 4.3. BDNF in neurological diseases. The chapter needs to be expanded. Ischemia should be discussed.

5) The role of Ca2+ ions in the mechanisms of action of BDNF is completely overlooked.

6) Figures illustrating the cytoprotective mechanisms of BDNF through activation of the immune system are needed.

Author Response

Reviewer #3 reports:

1) Line 137 - sentence broken

2) Express and are able to release BDNF from neurons

3) Figure 2 should also reflect the mechanism of action of pro-BDNF

4) 4.3. BDNF in neurological diseases. The chapter needs to be expanded. Ischemia should be discussed.

5) The role of Ca2+ ions in the mechanisms of action of BDNF is completely overlooked.

6) Figures illustrating the cytoprotective mechanisms of BDNF through activation of the immune system are needed.

We thank the Referee for her/his important suggestions and to raise crucial points. Accordingly, we have made the changes suggested in order to improve the manuscript.

1) We corrected the mistake.

2) We thank the referee for his/her comment, but unfortunately, we have not understood the question.

3) We modified the Figure 2 accordingly and consequently modified the legend.

4) We thank the referee for his/her suggestion, and accordingly, we added the following sentence about the important role of BDNF in cerebral ischemia: “BDNF also plays a role in cerebral ischemia. In animal models, ischemic brain injury is accompanied by increased expression and levels of BDNF, which can play a role in recovery and plasticity. Pretreatment with intraventricular BDNF infusion showed to reduce infarct size after focal cerebral ischemia in rats. In humans, serum BDNF levels were lower in the acute phases of ischemic stroke, suggesting that lower levels of BDNF were associated with a higher likelihood of cerebrovascular accidents”.

5) One of the pathways leads to the hydrolysis of phosphatidylinositol bisphosphate (PIP2) by phospholipase C gamma (PLCγ), which results in the formation of IP3 and subsequent mobilization of intracellular Ca2+. BDNF showed to increase Ca2+ levels in the cortex and hippocampus. Sustained intracellular Ca2 + elevation activated by BDNF is thought to participate in neuronal survival and may contribute to the onset of long-term potentiation.

6) As suggested, we have provided a figure on BDNF and the immune system.

Round 2

Reviewer 1 Report

Dear Authors,

I think that the reviewed version of the manuscript adequately addresses the points I raised, and can be now accepted for publication in IJMS.

Author Response

We thank the referee. 

Reviewer 3 Report

The authors took into account all my comments. However, the possibility of BDNF secretion (release) by neurons needs to be discussed.

Author Response

We thank the referee for his/her suggestion. Accordingly, we added the following sentence to the manuscript marked in red font “Neurons are the major cellular source of BDNF. It is mainly expressed in the hippocampus, cortex, amygdala, striatum and cerebellum in rodents and humans”.

Round 3

Reviewer 3 Report

my comments are taken into account. article can be accepted for publication